# Measuring entanglement entropy and its topological signature for phononic systems

Zhi-Kang Lin[1,5], Yao Zhou[2,5], Bin Jiang[3,5], Bing-Quan Wu[1,5], Li-Mei Chen[2], Xiao-Yu Liu[1], Li-Wei Wang[1], Peng Ye [2]✉ & Jian-Hua Jiang [1,3,4]✉

Entanglement entropy is a fundamental concept with rising importance in various fields ranging from quantum information science, black holes to materials science. In complex materials and systems, entanglement entropy provides insight into the collective degrees of freedom that underlie the systems' complex behaviours. As well-known predictions, the entanglement entropy exhibits area laws for systems with gapped excitations, whereas it follows the Gioev-Klich-Widom scaling law in gapless fermion systems. However, many of these fundamental predictions have not yet been confirmed in experiments due to the difficulties in measuring entanglement entropy in physical systems. Here, we report the experimental verification of the above predictions by probing the nonlocal correlations in phononic systems. We obtain the entanglement entropy and entanglement spectrum for phononic systems with the fermion filling analog. With these measurements, we verify the Gioev-Klich-Widom scaling law. We further observe the salient signatures of topological phases in entanglement entropy and entanglement spectrum.

Information, entropy, and entanglement have been found to play important roles in a diverse range of disciplines, including quantum information science, condensed matter physics, and nonequilibrium physics[1-5]. Starting from fundamental questions such as whether a subsystem contains enough information to describe its own physics and properties, quantum entanglement and nonlocal correlations have been identified as key quantities underlying fundamental physics and many intriguing emergent phenomena[1-5]. Entanglement entropy, a quantitative measure of quantum entanglement and nonlocal correlations, has become a powerful tool in the study of emergent phases of matter, properties of complex systems, and quantum criticalities beyond the conventional Landau-Ginzburg paradigm.

For instance, in many-body systems with gapped collective excitations, the entanglement entropy, i.e., the entropy of the quantum many-body ground state of a subsystem, is proportional to the area of its boundary[2,3,6]. This well-known "area law", i.e., $S \sim L^{d-1}$ where $L$ is the

subsystem's linear size and $d$ is its spatial dimension, originates from the nonlocal correlations in the system (Fig. 1a): Area law emerges because the nonlocal correlations decay exponentially with a short correlation length in such systems. The entanglement entropy is thus limited by the correlation length and the area of the subsystem's boundary (Fig. 1b). In contrast, gapless systems have long-range correlations, and the area law is hence violated (Fig. 1c)[2,3]. Field theories predict that the entanglement entropy has a logarithmic scaling law in one-dimensional (1D) gapless systems, $S \sim \log L$[7,8]. In $d$-dimensional fermion systems, according to the Gioev-Klich-Widom law, entanglement entropy has a logarithmic scaling, $S \sim L^{d-1} \log L$, which is quite general for fermions with a Fermi surface of $(d-1)$-dimensions[9-11]. This prediction, even in the free-particle regime, is highly nontrivial and has been discussed in many pioneering works[12-19]. Furthermore, it was proposed that the entanglement spectrum, the spectrum of the reduced density matrix of the subsystem, gives salient features such as

[1]School of Physical Science and Technology & Collaborative Innovation Center of Suzhou Nano Science and Technology, Soochow University, 1 Shizi Street, 215006 Suzhou, China. [2]Guangdong Provincial Key Laboratory of Magnetoelectric Physics and Devices, State Key Laboratory of Optoelectronic Materials and Technologies, and School of Physics, Sun Yat-sen University, 510275 Guangzhou, China. [3]Suzhou Institute for Advanced Research, University of Science and Technology of China, 215123 Suzhou, China. [4]School of Physical Sciences, University of Science and Technology of China, 230026 Hefei, China. [5]These authors contributed equally: Zhi-Kang Lin, Yao Zhou, Bin Jiang, Bing-Quan Wu. ✉e-mail: yepeng5@mail.sysu.edu.cn; jhjiang3@ustc.edu.cn

the entanglement gap and the in-gap entanglement spectrum that are connected to the topological band gap and the edge states spectrum (Fig. 1d)[20–26].

Despite the fundamental importance of the above predictions, much of them, however, have not yet been confirmed in experiments, with the only exception that the area laws were observed recently[27–29], which is mainly because measuring entanglement or nonlocal correlation is challenging in many systems[27–31]. Here, we circumvent such difficulty by characterizing the nonlocal correlation of phonons in 1D and two-dimensional (2D) phononic crystals and then obtain the entanglement entropy and entanglement spectrum from such non-local correlation. In this way, we verify the Gioev-Klich-Widom scaling law[9–11,19] by examining the scaling behaviors of entanglement entropy for 1D and 2D phononic systems with various dispersions. Our progress unveils a versatile platform for testing the fundamental properties of entanglement entropy. We further observe the salient signatures of phononic topological phases and their transitions in the entanglement spectrum and entanglement entropy. As entanglement spectrum and entanglement entropy provide quite general diagnosis of topological phases[20,21,32–34], our work opens a new approach toward experimental identification of topological phases and phase transitions without relying on the bulk-boundary correspondence.

## Results

### Measuring phononic nonlocal correlations and entanglement entropy

We illustrate the measurement protocol with phonon systems in finite 1D phononic crystals, as schematically shown in Fig. 2a. The phononic crystals have two identical cylindrical acoustic cavities in each unit-cell where the intra-unit-cell and inter-unit-cell couplings are controlled, respectively, by the radii of the connecting tubes, $r_1$ and $r_2$, mimicking the 1D Su-Schrieffer-Heeger model[34]. The details of the design and fabrication of the phononic crystals are presented in Supplementary Note 1. We first measure the pump-probe response of the phononic crystal by exciting acoustic waves at one cavity with a tiny speaker and detecting the acoustic signal at another cavity with a tiny microphone (Fig. 2a, b). We measure the pump-probe response for many configurations and analyze the data with a vector network analyzer (Keysight E5061B). After Fourier transformations, the measured position- and time-dependent response is converted into the response as a function of the wavevector and frequency. Such a response function is proportional to the single-particle Green's function of phonons which can then be spectrally decomposed to yield the phononic dispersion and Bloch wavefunctions via the singular value decomposition of the response tensor at resonance conditions (see Methods and Supple-

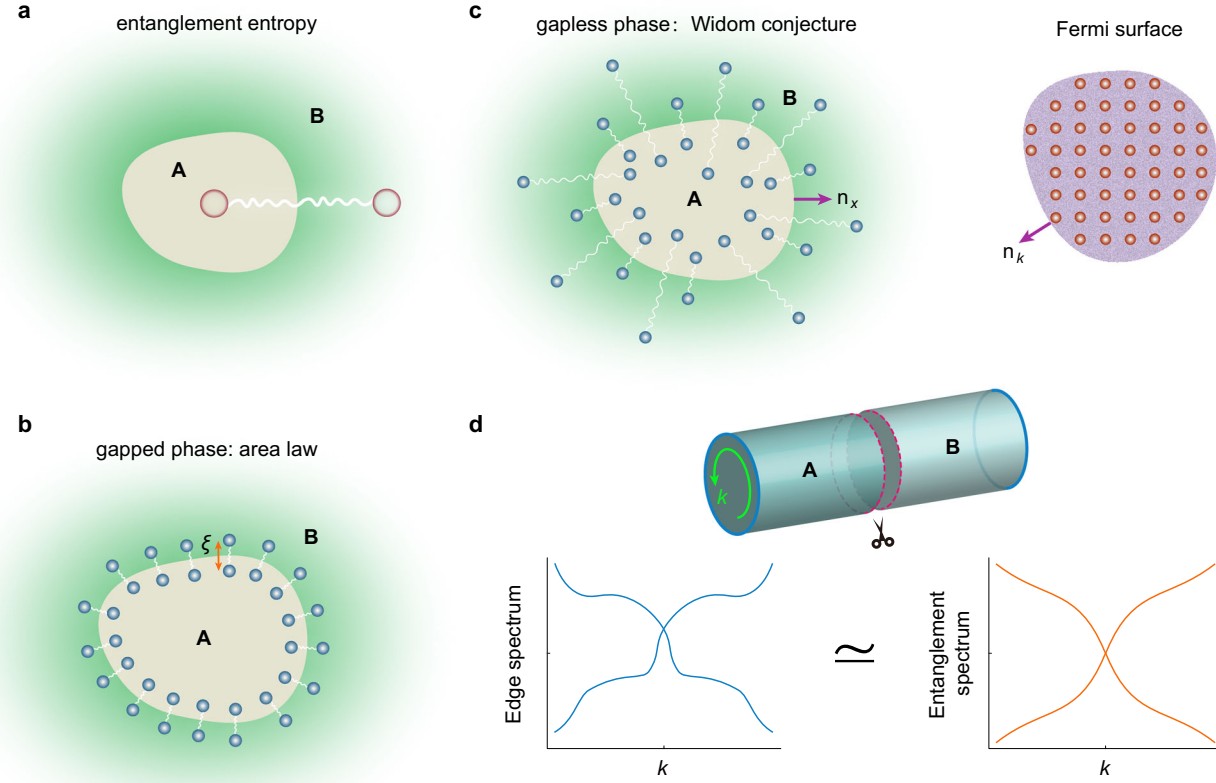

**Fig. 1 | Entanglement entropy, area law, and the Gioev-Klich-Widom scaling. a** Entanglement entropy of subsystem A quantifies the entanglement or nonlocal correlation (depicted by the wavy line) between subsystem A and the remaining subsystem B. **b** Area law of the entanglement entropy emerges in gapped phases where the correlation length $\xi$ is much smaller than the other length scales. Thus, all the correlations between A and B are around the boundary of subsystem A within a region of thickness $\xi$. **c** The Gioev-Klich-Widom law predicts a logarithmic scaling of entanglement entropy for gapless phases with a finite Fermi surface. **d** An emergent topological correspondence principle in the entanglement spectrum: the edge spectrum in the bulk band gap is connected to the entanglement spectrum by adiabatic principles. The inset illustrates the cutting of a cylinder surface (i.e., the 2D system with periodic boundary conditions along one direction) into two halves (A and B regions) to obtain the entanglement spectrum and entanglement entropy. The wavevector along the edge boundary is depicted by the green arrow.

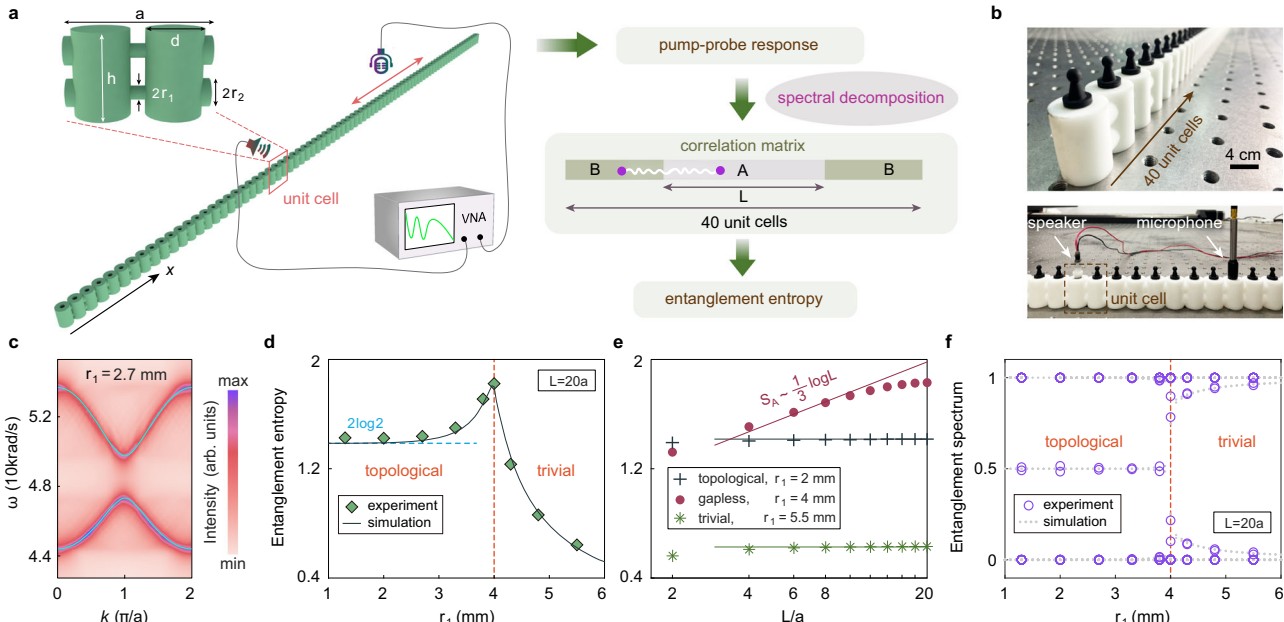

**Fig. 2 | Entanglement entropy and entanglement spectrum in 1D phononic systems. a** Measuring entanglement entropy in a 1D phononic crystal via the following procedure: First, measure the acoustic pump-probe response of the system with a speaker, a microphone, and a vector network analyzer (VNA). Second, extract the phononic dispersion and eigenstates by spectral decomposition of the pump-probe response. Third, construct the correlation matrix for subsystem A using the phononic dispersion and eigenstates. Fourth, determine the entanglement entropy from the correlation matrix. The upper-left inset gives the structure of a unit-cell of the phononic crystal. **b** Photograph of the 1D phononic crystal (top) and the measurement setup with a speaker as the source and a microphone as the detector (down). The black plugs seal the cavities when they are not used for

excitation or detection. **c** Phononic dispersion extracted from the pump-probe response. Cyan curves give the phononic dispersion from the full-wave simulation. **d** Entanglement entropy of the 1D phononic crystal versus the radius $r_1$ (representing the intra-unit-cell coupling). The contributions of all valence band states are considered with the fermion-filling analog. **e** Entanglement entropy versus the subsystem size $L$. Experimental data are represented by the points, while the lines give the area law or the logarithmic law. **f** Entanglement spectrum versus the radius $r_1$. Points represent the experimental data, while the dotted lines give the simulation results without dissipation. Orange dashed lines in **d** and **f** label the topological transition. Geometry parameters are: $a = 40$ mm, $h = 24$ mm, $d = 16$ mm, and $r_2 = 4$ mm.

mentary Note 2). An example of the extracted phononic dispersion is shown in Fig. 2c which agrees excellently with the phonon dispersion calculated from the full-wave simulation. With those extracted phonon dispersion and wavefunctions, we construct the two-point phonon correlation function $C_{\alpha\beta}(i,j,\omega)$ where $i$ and $j$ ($\alpha$ and $\beta$) denote the unit-cell (sublattice) indices of the two points and $\omega$ is the phonon's angular frequency. From the two-point correlation functions in the subsystem A, $C^A_{\alpha\beta}(i,j,\omega)$, by exhausting all pairs of points, one can determine the correlation matrix for the subsystem with the fermion-filling analog, $C^A_{\alpha\beta}(i,j) = \int_{\omega_b}^{\omega_F} d\omega C^A_{\alpha\beta}(i,j,\omega)$, where $\omega_F$ is the highest frequency of the "filled states" and $\omega_b$ is the lower cutoff frequency of the relevant bands (i.e., considering an analog of fermionic filling from $\omega_b$ to $\omega_F$). In the fermionic counterpart, the above gives the exact definition of the correlation matrix for the subsystem A with fermions filling all states below the Fermi energy $E_F = \hbar\omega_F$. Therefore, the equifrequency contour at $\omega_F$ gives the "Fermi surface". The above analog is exact because in the free-particle limit, single-particle fermion correlation functions can be exactly connected to the single-particle boson correlation functions. To fully establish the analog with free fermion systems so that we can test the Gioev-Klich-Widom scaling law and the topological entanglement spectrum, we determine the correlation matrix for the idealized systems without damping (see Methods).

The entanglement entropy can in principle be obtained from the reduced density matrix for the subsystem $\rho_A$ with the fermion-filling analog, or equivalently, from the eigenvalues of the correlation matrix $C^A$ as[35,36]

$$S_A = -\text{Tr}(\rho_A \log \rho_A) = -\sum_n \left[\varepsilon_n \log \varepsilon_n + (1 - \varepsilon_n) \log(1 - \varepsilon_n)\right] \quad (1)$$

where $\varepsilon_n$ are the eigenvalues of the correlation matrix $C^A$ which is termed as the entanglement spectrum. The equivalence of these two definitions is presented in Supplementary Note 3.

## Entanglement entropy and entanglement spectrum in 1D phononic systems

In experiments, we use 1D phononic crystals fabricated by the 3D printing technology with 40 unit cells and a lattice constant of $a = 40$ mm. With $r_2$ fixed to 4 mm and $r_1$ going from 1 mm to 6 mm, the phononic dispersion can be tuned from a topological band gap to a Dirac node, and then to a trivial band gap. Following the above procedure, we measure the entanglement entropy and entanglement spectrum in 1D phononic crystals with various $r_1$ by considering the filling of all states in the valence band. We find that the entanglement entropy exhibits notable changes by tuning $r_1$: The entanglement entropy has a peak at $r_1 = 4$ mm (Fig. 2d) which corresponds to a topological transition of the phononic system. This behavior is known as a remarkable feature of entanglement entropy at phase transitions which makes it invaluable in the study of phase transitions[32–34]. Moreover, in the topological gapped phase, the entanglement entropy is close to a constant $S_A \cong 2\log 2$—a salient signature of the 1D topological Su-Schrieffer-Heeger model (termed as the topological entanglement entropy)[20–26,34]. Unlike the near constant behavior in the topological phase, the entanglement entropy decreases quickly with increasing $r_1$ in the trivial gapped phase, which is an indirect manifestation of the area law: The increase of the trivial phonon band gap with increasing $r_1$ suppresses the correlation length and, hence, the entanglement entropy as well.

To verify the logarithmic scaling law for the gapless phase and the area law for the gapped phases in 1D, we study the dependence of the

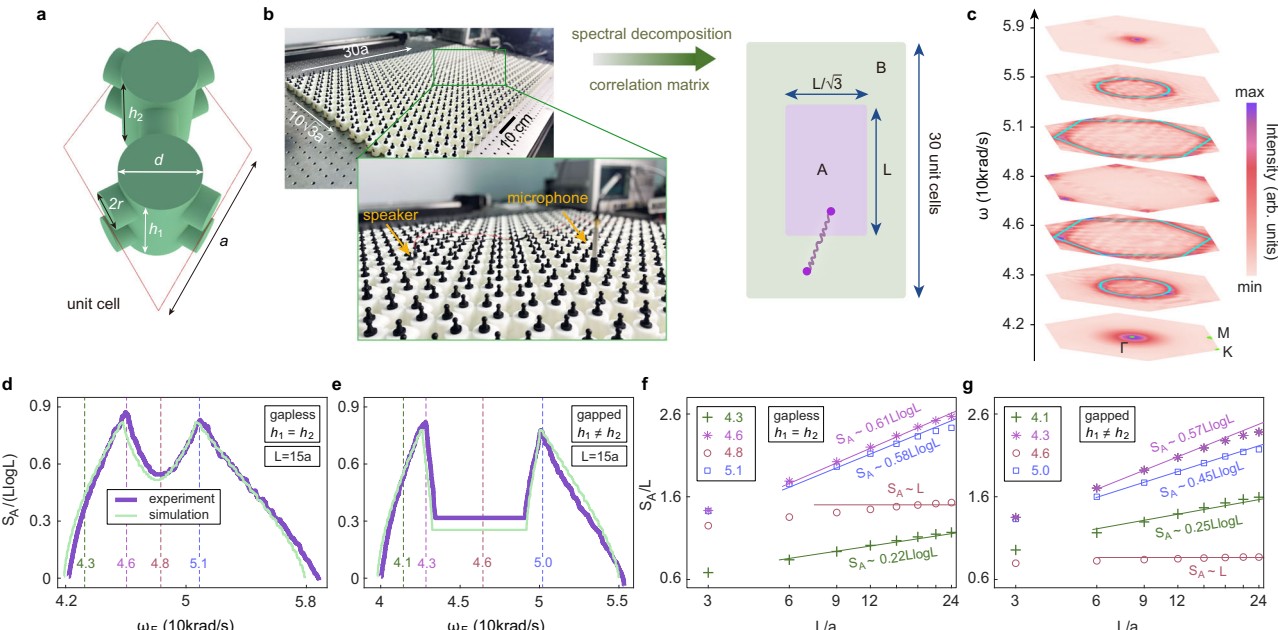

**Fig. 3 | Measuring entanglement entropy in 2D phononic systems. a** Schematic of the unit-cell structure of the 2D honeycomb phononic crystals. Geometry parameters: $a = 40mm$, $d = 16mm$, $r = 4mm$, $h_1 = 24mm$, and $h_2 = 24mm$ (28 mm) for the gapless (gapped) phase. **b** Photograph of a phononic crystal sample. The speaker and microphone used in pump-probe measurement are shown in the zoom-in image. From the same procedure as in Fig. 2, after measuring the pump-probe response, we perform the spectral decomposition, construct the correlation matrix of subsystem A, and obtain the entanglement entropy. **c** Measured phonon

equifrequency contours. Cyan curves represent the results from the full-wave simulation. **d**, **e** Measured entanglement entropy versus the "Fermi energy" $\omega_F$ for the gapless (**d**) and gapped (**e**) phases. **f**, **g** Measured entanglement entropy versus the subsystem's size $L$ for the gapless (**f**) and gapped (**g**) phases. Points represent the experimental data, while the lines give the scaling relations according to the Gioev-Klich-Widom law or the area law. The numerical prefactors in the scaling relations are calculated according to Eq. (2).

entanglement entropy on the subsystem's length, $L$. Area law states that in 1D the entanglement entropy is determined by the correlation length and is independent of $L$[2,3,6]. The experimental results in Fig. 2e indeed confirm such a dependence for both the topological and trivial gapped phases. For the gapless Dirac-node phase, conformal field theories predict the logarithmic scaling law[7,8], $S_A \cong \frac{1}{3} \log L$, which is well confirmed in Fig. 2e. The consistency with theory and simulations justifies the experimental verification of the fundamental scaling laws of entanglement entropy in 1D systems.

We further study the entanglement spectrum and its manifestation of topological phases and transitions. We observe in Fig. 2f that the entanglement spectrum experiences gap closing at the topological transition at $r_1 = 4$ mm. In the topological phase, there are in-gap entanglement spectrum near 0.5, whereas in the trivial phase, there is no such feature. Such doubly degenerate 0.5 spectrum is tied to the topological edge zero modes of the entanglement Hamiltonian and is the origin of the 2log2 topological entanglement entropy[21–24,33,34]. It is worth mentioning that the manifestation of topological transition in the entanglement spectrum is quite abrupt and thus can serve as a prominent signature of the topological transition. These results indicate that entanglement entropy and entanglement spectrum give a new probe of topological phases and topological transitions. Remarkably, this probe does not rely on the bulk-edge correspondence. Examples are presented in Supplementary Note 6 via a modified Su-Schrieffer-Heeger model without chiral symmetry where the usual bulk-boundary correspondence fails but entanglement entropy and entanglement spectrum still give reliable identification of the topological phase and the topological transition. Therefore, entanglement entropy and entanglement spectrum provide information of the intrinsic topology with no dependence on the chiral symmetry or the edge states. In fact, we characterized the band topology via pump-probe measurements in the bulk region instead of at the edges. Moreover, the detection of topological phases and topological

transitions via the entanglement entropy and entanglement spectrum can be achieved in finite systems as small as a dozen cells with subsystem sizes as small as a few unit cells (see Supplementary Note 7). Lastly, the location and geometry of subsystem A do not affect the main properties of entanglement entropy and entanglement spectrum, although they may modify the quantitative behaviors (see Supplementary Note 6). These properties indicate that the approach adopted here is useful in characterizing topological phases in an unconventional way, particularly for the fragile topological phases where the bulk-boundary correspondence is unstable[37].

## Entanglement entropy and entanglement spectrum in 2D phononic systems

In 2D phononic systems, the scaling behavior of the entanglement entropy can be explored with rich Fermi surface geometry. Here, our phononic crystals are based on honeycomb lattices with two cylindrical acoustic cavities in each unit-cell (Fig. 3a). The two cavities have the same diameter $d$, but their heights, $h_1$ and $h_2$, can be tuned. For $h_1 = h_2$, the system is a phononic graphene with Dirac nodes at the $K$ and $K'$ points. When $h_1 \neq h_2$, the Dirac nodes are gapped due to inversion symmetry breaking. For both cases, we fabricate one rectangular sample with a size of $30a \times 10\sqrt{3}a$ (the sample has zigzag ($30a$) and armchair ($10\sqrt{3}a$) edge boundaries; see Fig. 3b). Following the same procedure as in 1D, we measure the pump-probe response, perform the spectral decomposition, construct the correlation matrix, and lastly determine the entanglement entropy and entanglement spectrum.

Figure 3c presents the measured phonon equifrequency contours which agree well with the equifrequency contours from the full-wave simulation, indicating the success of the spectral decomposition of the pump-probe response. With increasing frequency, the equifrequency contour evolves from a ring at the Brillouin zone center, to a hexagon, then to two Dirac points at the $K$ and $K'$ points, and then reverse back.

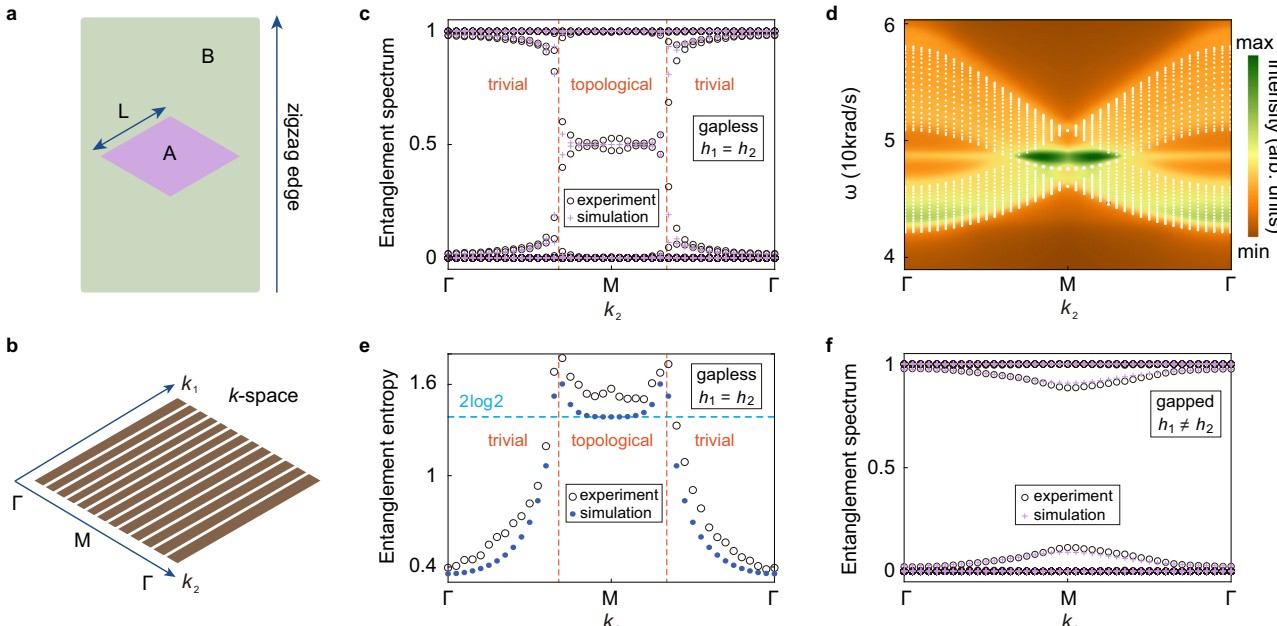

**Fig. 4 | Entanglement spectrum and topology in 2D honeycomb phononic crystals. a** To obtain the $k_2$-dependent entanglement spectrum, we construct the correlation matrix of a rhombus subsystem A with the side length $L = Sa$, which has zigzag-type boundaries. **b** Schematic of the $k$-space in the rhombic first Brillouin zone. For each $k_2$, we construct the correlation matrix of the rhombus subsystem A for half-filling and obtain the entanglement spectrum. **c** Entanglement spectrum versus $k_2$ for the gapless phononic graphene. Both experimental data and the results from full-wave simulations without the dissipation effect are presented. **d** Measured phononic dispersion on the zigzag edge obtained from analyzing the pump-probe response around a zigzag edge boundary. The colormap gives the intensity of the detected acoustic signal after Fourier transformation for the pump-probe measurements along an edge boundary, i.e., it gives the spectral intensity along the edge boundary at various wavevectors and frequencies. **e** Entanglement entropy from both experiments and simulations versus $k_2$. Here the full-wave simulation also does not include the dissipation effect. **f** Entanglement spectrum versus $k_2$ for the gapped phase with broken inversion symmetry. Orange dashed lines in **c** and **e** denote the projection of the two Dirac points in the $k_2$ axis. Geometry parameters are the same as in Fig. 3.

Such an evolution of the "Fermi surface" has profound effects on the entanglement entropy. According to the Gioev-Klich-Widom law[9–11,19], the entanglement entropy of $d$-dimensional gapless fermions with a $(d$-1)-dimensional Fermi surface has the following scaling behavior[10,11]

$$S_A \cong \frac{L^{d-1}\log L}{12(2\pi)^{d-1}} \int\!\!\int |\boldsymbol{n}_x \cdot \boldsymbol{n}_k| dA_x dA_k, \qquad (2)$$

where $dA_x$ and $dA_k$ are the infinitesimal areas at the boundaries of the subsystem and the Fermi surface, respectively. $\boldsymbol{n}_x$ and $\boldsymbol{n}_k$ are the unit normal vectors at these boundaries (Fig. 1c). The above integral is defined in a scaled way such that it gives only a numerical factor. More details are presented in Supplementary Note 4.

Indeed, the experimental results show that the entanglement entropy has a strong dependence on the "Fermi energy", i.e., $\omega_F$. For the phononic graphene with $h_1 = h_2$, Fig. 3d shows that entanglement entropy grows from zero in the empty band limit to a peak value at the valence band hexagonal Fermi surface at 46 krad/s, and then reaches a local minimum at the Dirac nodes at 48 krad/s. These observations are consistent with Eq. (2) which predicts the local maximum (minimum) of the entanglement entropy when the size of the Fermi surface reaches a local maximum (minimum). The entanglement entropy further develops another peak at the conduction band hexagonal Fermi surface at 51 krad/s, and finally goes back to zero in the all-bands filling limit. These results indicate that the entanglement entropy increases (decreases) with increasing (decreasing) Fermi surface which is consistent with the Gioev-Klich-Widom law. It is worth mentioning that the all-bands filling limit corresponds to vanishing correlation length and hence zero entanglement entropy. Furthermore, for the gapped case with $h_1 \neq h_2$, Fig. 3e shows similar behaviors, except for the emergence of a frequency window with no change in the entanglement entropy which corresponds to the band gap where no states exist. For all these

cases, the experimental results agree well with the entanglement entropy obtained from the full-wave simulations based on the acoustic wave equations (see details in Supplementary Note 5). Small deviations may be due to that dissipation effects are not included in these simulations.

As shown in Fig. 3f, g, the scaling behaviors of the entanglement entropy in 2D are quite different for finite and Dirac-node Fermi surfaces. For finite Fermi surfaces, the entanglement entropy follows the logarithmic scaling predicted by the Gioev-Klich-Widom law[9–11,19], i.e., $S_A \sim L \log L$. To fully verify the Gioev-Klich-Widom scaling law, for each case with a finite Fermi surface in Fig. 3f, g, we calculate the scaling relation of the entanglement entropy according to Eq. (2). These scaling relations are presented as lines in Fig. 3f, g, while the experimental data are shown as points. The consistency between the experimental points and the theoretical lines is notable for all cases with finite Fermi surfaces: Not only the logarithmic scaling behavior but also the numerical prefactors in the scaling relations agree well with the experimental data (Fig. 3f, g). Such consistency is particularly excellent for moderate $L$, while for very large $L$ (i.e., $L$ comparable with the size of the system), the finite-size effect can cause deviation between the experiments and the theory (see details in Supplementary Note 4). The above results provide a sound verification of the Gioev-Klich-Widom scaling law.

It is worth mentioning that Fermi surfaces with Dirac nodes correspond to intriguing cases. In 1D, the codimension of the Dirac-node Fermi surface is 1, and conformal field theories predict a scaling law of $S_A \cong \frac{c_L + c_R}{6} \log L$ where $c_L = c_R = 1$ are, respectively, the central charges of left- and right-moving modes[7,8] (see Supplementary Note 6). The experimental results in Fig. 2e are consistent with this prediction. In contrast, in 2D the codimension of the Dirac-node Fermi surface is 2, leading to the same scaling law as in gapped systems[38], i.e., the area law $S_A \sim L$. However, in finite experimental systems, the genuine scaling is

in between the area law and the logarithmic law, as seen in Fig. 3f. In comparison, when the Fermi energy is in the band gap, the entanglement entropy follows a clear area law behavior in experiments (Fig. 3g).

We now demonstrate from experiments that the entanglement spectrum of 2D phononic crystals gives a direct manifestation of the band topology. We study the entanglement spectrum for a rhombus subsystem with fixed $k_2$ (see Fig. 4a, b) which gives an effective 1D phononic system with $k_2$ being a parameter. The entanglement spectrum is extracted from the measured pump-probe responses using our standard procedure except that the periodic boundary conditions are imposed in the $k_2$ direction for a rhombus subsystem (Fig. 4a). We consider the half-filling case where all the valence band states are filled. For the phononic graphene phase, the measured entanglement spectrum shows a gap closing at the Dirac points and the emergence of the in-gap spectrum around 0.5 when $k_2$ is between the projection of the two Dirac points (Fig. 4c). This behavior, which also agrees well with the full-wave simulation, indicates the nontrivial topology when $k_2$ is between the projection of the two Dirac points[20–24]. Indeed, the zigzag edge states emerge in this region as shown in Fig. 4d (see Supplementary Note 7 for details of the edge dispersion measurements), which is consistent with the known property of graphene[39]. These findings directly manifest the relationship between the entanglement spectrum of the bulk and the dispersion of the physical edge states—an alternative form of the bulk-edge correspondence predicted recently[20–24]. The band topology can also be revealed by the $k_2$-dependent entanglement entropy (Fig. 4e): In between the projection of the two Dirac points, the entanglement entropy is about 2log2—a signature of nontrivial topology[32–34]. Outside this $k_2$ range, the entanglement entropy decreases quickly with the increasing band gap, agreeing with the properties of the trivial band gap as in Fig. 2d. More discussions on the topology and its manifestation on the $k_2$-dependent Zak phase from both experiments and simulations as an alternative characterization[32] of the band topology can be found in Supplementary Note 8. Finally, we remark that the band topology discussed above is essentially protected by the inversion symmetry. When the inversion symmetry is broken, the aforementioned topology should break down. As shown in Fig. 4f, the measured and simulated entanglement spectrum has no gap-closing feature and thus no indication of nontrivial topology. Here, such behavior is caused by the breaking of the inversion symmetry by setting $h_1 \neq h_2$. In this inversion symmetry broken phase, there is a complete phononic band gap, yet the $k_2$-dependent entanglement spectrum is trivial.

## Discussion

We achieve here the experimental verification of the Gioev-Klich-Widom scaling law of entanglement entropy for gapless systems[9–11,19]. We further demonstrate the manifestation of topological phases in the measured entanglement entropy and entanglement spectrum. These observations confirm the fundamental properties of entanglement entropy that have been anticipated for a long time. Our study opens a new pathway to the experimental investigation of entanglement and topology based on phononic systems. In theory, entanglement entropy and entanglement spectrum offer a powerful probe of various topological phases and their transitions, ranging from integer and fractional quantum Hall states to topological insulators, fragile and higher-order topological phases[20–25,32–34,37,40]. Our work thus provides a new experimental approach for probing topological phases and their transitions without relying on bulk-boundary correspondence. Furthermore, the approach developed here can be generalized to other systems such as cold atom gases to reveal the signatures of rich Fermi surface transitions (e.g., Lifshitz transitions) in entanglement entropy[18]. Finally, considering the advantages in realizing non-Hermitian effects in phononic systems[41], our work also serves as an important leap to studying entanglement entropy in non-Hermitian systems, where many intriguing predictions such as the transition

between the area law and the volume law are yet to be confirmed in experiments[42–46].

## Methods

### Experiments

For both 1D and 2D phononic crystals, we use the same procedure to measure the entanglement entropy, the phononic dispersion, and the entanglement spectrum. This procedure starts by measuring the pump-probe response in the phononic crystal. In such a measurement, a tiny speaker, serving as the source, is inserted into a cavity in the sample to excite the acoustic waves. In our measurements, the excitation frequency sweeps from 40 krad/s to 60 krad/s with a step of about 50 rad/s. A tiny microphone, connected to the vector network analyzer (Keysight E5061B), is inserted into each cavity other than the cavity with the speaker to detect the acoustic signal. By moving the source and the detector to traverse all cavities, we can, in principle, measure the pump-probe response for all possible configurations. The detailed results of the measured pump-probe response are shown in Supplementary Note 2.

From the measured pump-probe response, we extract the phonon dispersion and wavefunctions via a spectral decomposition method. For instance, to extract the bulk phonon dispersion and wavefunctions in 2D phononic crystals, we measure the pump-probe response away from the sample boundaries $\chi_{\alpha\beta}(i,j,t,t')$ where $i$ and $\alpha$ ($j$ and $\beta$) are the unit-cell and sublattice indices of the cavity where the source (detector) resides, $t$ ($t'$) is the time when the acoustic wave is launched (detected). For both the 1D lattices in Fig. 2 and the honeycomb lattices in Figs. 3 and 4, the sublattice indices $\alpha,\beta = 1,2$.

It is worth mentioning that the pump-probe response in real space has phase ambiguity that cannot be fixed in experiments due to the arbitrary initial phase. Therefore, $\chi_{\alpha\beta}(i,j,t,t')$ is not suitable for the evaluation of the correlation function or the correlation matrix which are key quantities for the evaluation of the entanglement entropy and entanglement spectrum from experimental data. It should also be noted that the correlation matrix $C_{i,j}^{A}$ is made of equal-time correlation functions which cannot be directly determined from the measured pump-probe response (The measured pump-probe responses are essentially connected to the single-particle retarded Green's function of phonons).

We need to invent a method to avoid such difficulties. For this purpose, we Fourier transform the pump-probe signals from position-time space to wavevector-frequency space which leads to the response tensor $\chi_{\alpha\beta}(\mathbf{k},\omega)$. This response tensor is proportional to the single-particle retarded Green's function of phonons, i.e.,

$$\chi_{\alpha\beta}(\mathbf{k},\omega) \propto \sum_{n\mathbf{k}} \frac{u_{n\mathbf{k}}^{*}(\alpha)u_{n\mathbf{k}}(\beta)}{\omega - (\omega_{n\mathbf{k}} + i\gamma_{n\mathbf{k}})}, \tag{3}$$

where $u_{n\mathbf{k}}$ is the periodic part of the phonon Bloch wavefunction ($n$ is the band index and $\mathbf{k}$ is the wavevector) in the sublattice site space, i.e., it is a normalized two-component vector. We first observe that the response intensity has the following properties:

$$P(\mathbf{k},\omega) = \sum_{\alpha,\beta} \chi_{\alpha\beta}^{*}(\mathbf{k},\omega)\chi_{\alpha\beta}(\mathbf{k},\omega) \propto \sum_{n\mathbf{k}} \frac{1}{(\omega - \omega_{n\mathbf{k}})^2 + \gamma_{n\mathbf{k}}^2}. \tag{4}$$

Here, we have used $\sum_{\alpha,\beta} u_{n\mathbf{k}}^{*}(\alpha)u_{n\mathbf{k}}(\beta)u_{n\mathbf{k}}(\alpha)u_{n\mathbf{k}}^{*}(\beta) = (\sum_{\alpha}|u_{n\mathbf{k}}(\alpha)|^2)(\sum_{\beta}|u_{n\mathbf{k}}(\beta)|^2) = 1$, i.e., the normalization condition, $\sum_{\alpha}|u_{n\mathbf{k}}(\alpha)|^2 = 1$. Therefore, the response intensity $P(\mathbf{k},\omega)$ consists of many Lorentzian functions and contains all the spectral information of the system. By fitting these Lorentzian peaks, we can obtain the phonon dispersion $\omega_{n\mathbf{k}}$ (i.e., the peak frequency of $P(\mathbf{k},\omega)$) and the damping parameter $\gamma_{n\mathbf{k}}$ (i.e., the Lorentzian broadening of the resonance peak). Excellent examples of such fitting are shown in Supplementary Fig. 6.

Another key observation is that when the frequency $\omega$ is in resonance with a peak, i.e., $\omega = \omega_{nk}$, the dominant contribution of the summation in Eq. (3) comes from the resonant band. We use a singular value decomposition (SVD) of the response tensor $\chi(\boldsymbol{k},\omega)$ to extract the dominant contribution, i.e.,

$$\chi(\boldsymbol{k},\omega_{nk}) = U(\boldsymbol{k},\omega_{nk})\Sigma(\boldsymbol{k},\omega_{nk})V^{\dagger}(\boldsymbol{k},\omega_{nk}). \qquad (5)$$

Here $U(\boldsymbol{k},\omega_{nk})$ and $V(\boldsymbol{k},\omega_{nk})$ contain the left and right eigenvectors, respectively, while $\Sigma(\boldsymbol{k},\omega_{nk})$ contains the eigenvalues. The SVD of the $2\times 2$ response tensor $\chi(\boldsymbol{k},\omega_{nk})$ gives both the largest eigenvalue and the corresponding left and right eigenvectors. From Eq. (3), one can see that the right eigenvector gives exactly the phonon's wavefunction $u_{nk}$, up to a phase redundancy (Note that all eigenvectors are normalized).

From the extracted phonon dispersion $\omega_{nk}$ and wavefunction $u_{nk}$, we can then construct the frequency-dependent correlation function $C^A(i,j,\omega)$ in the subsystem A,

$$C^A_{\alpha\beta}(i,j,\omega) = \sum_{nk} \delta(\omega - \omega_{nk})\psi^{*}_{nk}(i,\alpha)\psi_{nk}(j,\beta). \qquad (6)$$

Here, the two points $(i,\alpha)$ and $(j,\beta)$ exhaust all possible sites in the subsystem A. The delta function here counts the contribution on the angular frequency $\omega$ which is from the definition of the entanglement entropy[12–19,42–46]. The above summation over wavevector is performed on a $40\times 40$ mesh (40 mesh) in the Brillouin zone for 2D (1D) phononic crystal where the quantities $\psi_{nk}$ and $\omega_{nk}$ are determined and extracted from the pump-probe experimental data. With the frequency-dependent correlation function, we determine the correlation matrix and then the entanglement entropy and entanglement spectrum via Eq. (1).

We remark that the above approach is an effective approximation, for we assumed the lattice translation symmetry and periodic boundary conditions. However, for sufficiently large systems like the phononic crystals examined here, this approximation effectively captures the intrinsic properties of the bulk states such as the bulk phonon dispersion and wavefunctions when the acoustic source is placed at the center of the phononic crystals (The edge properties can also be obtained if the pump-probe is around the edge boundaries). Furthermore, without such an approximation, one must exhaust all possible pump-probe configurations in the measurements which is practically not feasible for 2D phononic crystals as it involves $1220 \times 1220$ pump-probe configurations in total. In addition, such brutal-force measurements may give only marginal corrections to the results obtained in this work. Lastly, we find that our method is not only efficient in extracting the equal-time correlation functions from the pump-probe experimental data but also effective in avoiding the intrinsic frequency dependences of the experimental instruments (especially the acoustic source and detector) which can be notable and detrimental as broad frequency range measurements are involved in this work (see Supplementary Note 2 for details).

### Simulations

All simulations of acoustic wave dynamics and calculation of acoustic phonon dispersions and wavefunctions were performed via using an acoustic pressure module of the commercial finite-element solver COMSOL Multiphysics. Due to the huge acoustic impedance mismatch between air and the photosensitive resins used in the 3D printing technology, the resins can be regarded as hard boundaries for the acoustic waves in the numerical simulations. Acoustic waves propagate in the air regions encapsulated by the resins. The mass density of air is 1.29 kg/m³ and the sound speed is 346 m/s at room temperatures (around 25 °C). The phononic dispersion and wavefunctions in the phononic crystals are calculated with the Floquet Bloch boundary conditions. To calculate the projected bands of the supercell with the zigzag edge boundaries, the system is set as having Floquet periodic

boundary conditions along the $k_2$ direction but closed boundary conditions along the other finite direction. In this work, the entanglement entropy and entanglement spectrum from the simulation means that they are obtained from the phonon correlation function and correlation matrix constructed from the simulated phonon dispersion and wavefunctions based on the acoustic wave equations.

## Data availability

All data needed to evaluate the conclusions in this paper are present in the manuscript and the Supplementary Information. Additional information is available from the corresponding authors through request.

## Code availability

We use the commercial software COMSOL MULTIPHYSICS to perform the finite-element acoustic wave simulations and calculations. Reasonable requests for the computation details can be addressed to the corresponding authors.

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

## Acknowledgements

J.H.J. thanks the support from the National Key R&D Program of China (2022YFA1404400), the National Natural Science Foundation of China under grant Nos. 12125504 and 12074281, the CAS Pioneer Hundred Talents Program, Gusu Leading Innovation Scientists Program of Suzhou City, and the Priority Academic Program Development (PAPD) of Jiangsu Higher Education Institutions. P.Y. was supported by the National Natural Science Foundation of China under the grant No. 12074438, the Guangdong Basic and Applied Basic Research Foundation under the grant No. 2020B1515120100, and the Open Project of Guangdong Provincial Key Laboratory of Magnetoelectric Physics and Devices under the grant No. 2022B1212010008.

## Author contributions

J.H.J. and P.Y. initiated and guided the research. P.Y., L.M.C., and Y.Z. established the theory. Z.K.L. and J.H.J. designed the phononic systems. Z.K.L. performed all the simulations. Z.K.L., B.J., B.Q.W., X.Y.L., and L.W.W. performed the experiments and data processing. All the authors contributed to the discussions of the results and the manuscript preparation. J.H.J., P.Y., Z.K.L., Y.Z., and L.M.C. wrote the manuscript.

## Competing interests

The authors declare no competing interests.
