## [Peer Review File · Nature Communications]

Measuring entanglement entropy and its topological signature for phononic systemsREVIEWER COMMENTS

Reviewer #1 (Remarks to the Author):

The paper "Experimental verification of Widom's conjecture and characterization of topology without bulk-boundary correspondence" presents an experiment where the phononic correlation function is measured and used to represent the correlation function of a free fermion system. I enjoyed the simplicity of the demonstration in comparison to the cold atom type experiments and think it would be a worthwhile publication with an appeal to the large community that studies entanglement. However, I believe the present draft requires some rewording in order to properly state the result.

In particular, the paper is not a verification of Widom's conjecture: the conjecture has nothing to do with entropy or fermions, rather it has to do with a rather broad class of traces of functions of Toeplitz operators, and in fact has been recently proven for a class of cases, and is more appropriately called the Widom-Sobolev formula.

A more appropriate claim would be a demonstration or verification of the Goev-Klich-Widom scaling of entanglement entropy for free fermions. Here, authors must be careful to be clear that - they do not directly measure many body entropy, rather they measure a correlation function, similarly, say to their ref [29]. Note, however, that in contrast to [29] their measurement is of a correlation matrix for an idealized system that is not physically their system.

Because the (frequency integrated) correlation function represents the equal time correlation function of idealized non-interacting fermions, they can use the experimental result to compute entropy for such an idealized system using the connection between fermion correlation matrix and entropy and to get to the scaling found in ref [10]. Note that [10] is based on using the Widom conjecture but obtained for a particular case of the conjecture, but outside its originally proposed range of validity and subsequent proof by Sobolev. In this sense the result is similar to the numerical verification in ref [15] etc.

Reviewer #2 (Remarks to the Author):

The authors have developed an acoustic system based on interconnected resonating

cavities to conduct experimental measurements of entanglement entropy, which serves as a quantitative measure of quantum entanglement and non-local correlation, facilitating the study of various phases of matter. This kind of acoustic system have been effectively used to demonstrate analog phenomena in condensed matter physics, such as topological insulators. More broadly, classical wave-based systems have emerged as powerful tools for showcasing phenomena that are experimentally challenging to access in quantum mechanics. While recent studies have observed area laws in optical systems, the uniqueness of this work lies not only in demonstrating non-local phonon correlation within 1D and 2D phononic crystals, but also in validating Widom's conjecture. The authors have provided direct evidence of topological phases through the measured entanglement entropy and entanglement spectrum. In light of these contributions, I believe this work merits publication in Nature Communications. I do have a couple of questions and suggestions: Could the authors elaborate on why the loss appears to have minimal impact on their experimental observations? This clarification would enhance the reader's understanding of the robustness of their system.

Although this platform serves as a formidable tool for illustrating entanglement entropy, could the authors offer insights into how their findings might advance the field of acoustics, potentially leading to practical applications?

Reviewer #3 (Remarks to the Author):

In this manuscript, the authors present phononic lattices in 1D and 2D and perform pump-probe measurements to obtain entanglement entropy and entanglement spectrum. Moreover, the authors claim to verify the Widom conjecture of entanglement entropy by checking the area law. The authors also propose that the entanglement spectrum can be used to probe topological phases of materials without relying on bulk-boundary correspondence.

The manuscript is well-written, and the experimental results are valid. The findings are appealing. However, I would like to clarify the following points before I can rightly assess its impact and significance:

Major concern:

Topological phase transitions in phononic crystals have been investigated extensively in recent years. The current study proposes a different route to probe such topological phases. However, the current version of the manuscript does not make it clear how this new route could unveil the topological phase of matter where the usual bulk-boundary correspondence fails or is difficult. If the authors can address this issue, the findings would have a much stronger appeal to the community.

Minor concerns/suggestions:

- In Fig. 2e, why does experimental data deviate from the area law for low L ? What do simulations predict?
- For low L , which is more practical in experimental settings, can a clear topological transition be predicated, as shown in Fig. 2d?
- How does the location of subsystem A influence the area law shown in Fig. 2e? For example, if one takes the same size of subsystem but closer to the boundary of the chain, would one expect any deviation?
- Does the value $2\log 2$ for the entanglement entropy depend on the fact the system has chiral or inversion symmetry? Or the existence of edge states is sufficient for one to arrive at this value? If the latter is true, how can one justify that the state is always topologically nontrivial?
- What explains the region of local minima in Fig. 3d? I believe this depends on the shape of the subsystem A . Therefore, the authors may want to discuss the effect of choosing different shapes of A .
- At what frequency Figs. 4c and 4e are plotted? I believe the edge spectrum has nonzero group velocity, shown in Fig. 4d. Therefore, it is not clear how the choice of frequency would influence the entanglement spectrum and entropy and prove the existence of edge states.

- The colormap in Fig. 4d needs an explanation.

Reply to Reviewer #1

Reviewer's comments: *The paper "Experimental verification of Widom's conjecture and characterization of topology without bulk-boundary correspondence" presents an experiment where the phononic correlation function is measured and used to represent the correlation function of a free fermion system. I enjoyed the simplicity of the demonstration in comparison to the cold atom type experiments and think it would be a worthwhile publication with an appeal to the large community that studies entanglement. However, I believe the present draft requires some rewording in order to properly state the result. In particular, the paper is not a verification of Widom's conjecture: the conjecture has nothing to do with entropy or fermions, rather it has to do with a rather broad class of traces of functions of Toeplitz operators, and in fact has been recently proven for a class of cases, and is more appropriately called the Widom-Sobolev formula. A more appropriate claim would be a demonstration or verification of the Gioev-Klich-Widom scaling of entanglement entropy for free fermions. Here, authors must be careful to be clear that - they do not directly measure many body entropy, rather they measure a correlation function, similarly, say to their ref [29]. Note, however, that in contrast to [29] their measurement is of a correlation matrix for an idealized system that is not physically their system. Because the (frequency integrated) correlation function represents the equal time correlation function of idealized non-interacting fermions, they can use the experimental result to compute entropy for such an idealized system using the connection between fermion correlation matrix and entropy and to get to the scaling found in ref [10]. Note that [10] is based on using the Widom conjecture but obtained for a particular case of the conjecture, but outside its originally proposed range of validity and subsequent proof by Sobolev. In this sense the result is similar to the numerical verification in ref [15] etc.*

Our reply: We thank the reviewer for his/her appreciation and comments of our work. According to the comments, we decided to change the title to "Measuring entanglement entropy and its topological signatures for phononic systems" which we believe better characterize the innovation in this work. In the revised main text, we also replace the "Widom conjecture" with the "Gioev-Klich-Widom scaling". In addition, we revised the main text so that it is clear in the introduction part that we obtain the entanglement entropy from the correlation matrix. We also corrected a few typos and revised the statements to clarify the underlying logic.

Reply to Reviewer #2

Reviewer's remarks: *The authors have developed an acoustic system based on interconnected resonating cavities to conduct experimental measurements of entanglement entropy, which serves as a quantitative measure of quantum entanglement and non-local correlation, facilitating the study of various phases of matter. This kind of acoustic system have been effectively used to demonstrate analog*

phenomena in condensed matter physics, such as topological insulators. More broadly, classical wave-based systems have emerged as powerful tools for showcasing phenomena that are experimentally challenging to access in quantum mechanics. While recent studies have observed area laws in optical systems, the uniqueness of this work lies not only in demonstrating non-local phonon correlation within 1D and 2D phononic crystals, but also in validating Widom's conjecture. The authors have provided direct evidence of topological phases through the measured entanglement entropy and entanglement spectrum. In light of these contributions, I believe this work merits publication in Nature Communications.

Our reply: We thank the reviewer for his/her appreciation and comments of our work.

Reviewer's comments: *I do have a couple of questions and suggestions: Could the authors elaborate on why the loss appears to have minimal impact on their experimental observations? This clarification would enhance the reader's understanding of the robustness of their system.*

Our reply: In our experiments, we measure the idealized correlation matrix for the phononic system with free fermion analog. That is, we extract the Bloch wavefunctions and their eigenfrequencies via pump-probe measurements, while the loss has been eliminated in constructing the correlation matrix. In this way, we establish an excellent analog of the entanglement entropy in free fermion systems in both 1D and 2D for both the gapless and gapped systems. Although the loss does have negative effects on the experiments, fortunately in our system it is negligible and nearly uniform. Therefore, it does not affect the experimental results, and our approach applies well to the phononic system studied and yields experimental results agreeing excellently with the theoretical calculations.

Reviewer's comments: *Although this platform serves as a formidable tool for illustrating entanglement entropy, could the authors offer insights into how their findings might advance the field of acoustics, potentially leading to practical applications?*

Our reply: We thank the reviewer for this inspiring question. From the acoustic aspect, there is indeed something that might be interesting. For instance, entanglement entropy is also an information tool that can be used to measure the information carried by the acoustic waves in a certain frequency range and a certain part of the system. In fact, entropic information tool has been used to analyze the acoustic signals in applications such as defect and damage monitoring in mechanical systems (e.g., railway systems and large-scale machines, see, e.g., *Journal of Sound and Vibration* **339**, 419-432 (2015), *Mechanical Systems and Signal Processing* **100**, 617-629 (2018)). In acoustic dynamics, the entanglement entropy can be used to monitor the information inflow and outflow carried by the acoustic waves which can be used to infer the global, collective properties of acoustic dynamics in complex systems.

Reply to Reviewer #3

Reviewer's remarks: *In this manuscript, the authors present phononic lattices in 1D and 2D and perform pump-probe measurements to obtain entanglement entropy and entanglement spectrum. Moreover, the authors claim to verify the Widom conjecture of entanglement entropy by checking the area law. The authors also propose that the entanglement spectrum can be used to probe topological phases of materials without relying on bulk-boundary correspondence. The manuscript is well-written, and the experimental results are valid. The findings are appealing.*

Our reply: We thank the reviewer for his/her appreciation and comments of our work.

Reviewer's comments: *However, I would like to clarify the following points before I can rightly assess its impact and significance: Major concern: Topological phase transitions in phononic crystals have been investigated extensively in recent years. The current study proposes a different route to probe such topological phases. However, the current version of the manuscript does not make it clear how this new route could unveil the topological phase of matter where the usual bulk-boundary correspondence fails or is difficult. If the authors can address this issue, the findings would have a much stronger appeal to the community.*

Our reply: Indeed, this is a point that must be addressed. In phononic systems with the time-reversal symmetry, the topology is often more fragile than the community has perceived. Take the simplest example of the phononic SSH model as an example. Here, the topological edge states must be protected by the chiral symmetry which is often absent in phononic crystals. Without the chiral symmetry, the topological edge states may not exist. In fact, in our systems the edge states disappear in the topological band gap due to the absence of the chiral symmetry [see Fig. R1 below]. In contrast, the entanglement entropy and entanglement spectrum provide faithful indication of the band topology in the absence of the chiral symmetry. Therefore, the approach demonstrated in this work gives a more robust measure of the band topology even when the usual bulk-boundary correspondence fails. We added these statements in the revised main text and the supplementary information.

Figure R1 | Topological transitions, edge spectrum and entanglement entropy. Phononic crystals studied in this work lacks the chiral symmetry due to inevitable next-nearest-neighbor couplings. Absence of the chiral symmetry leads to the breakdown of the usual bulk-boundary correspondence. **a**, Eigen-spectrum of finite phononic crystals (with 40 unit-cells) as a function of the radius of the intra-unit-cell coupling tube r_1 . In the topological phase ($r_1 < 4\text{mm}$), due to the absence of chiral symmetry, the edge states (labeled by the red curve) disappear for the topological phase with $2.4\text{mm} < r_1 < 4\text{mm}$, while they appear only for $r_1 < 2.4\text{mm}$. Here the other

geometry parameters are the same as in the main text. **b**, By changing the parameters of the cylindrical cavities as $h=50\text{mm}$ and $d=18\text{mm}$, the chiral symmetry is broken more severely, leading to the absence of the edge states in the topological phase in the whole range of $1\text{mm}<r_1<4\text{mm}$. For the cases in **b**, the topological phase can still be faithfully identified via the entanglement entropy and entanglement spectrum. **c**, Entanglement entropy as a function of the radius r_1 clearly indicates the topological transition at $r_1=4\text{mm}$ and gives the topological entanglement entropy of $2\log 2$ in large gap cases (i.e., small r_1 cases). **d**, Most importantly, the entanglement spectrum changes sharply at the topological transition. In the topological phase (almost for the whole parameter range with $r_1<4\text{mm}$), a gap and a branch at 0.5 emerges in the entanglement spectrum which is a clear and faithful indication of the topological phase even in the absence of the chiral symmetry. These results demonstrate clearly the power of the entanglement entropy and entanglement spectrum in identification of topological phases.

Reviewer's comments: *Minor concerns/suggestions: In Fig. 2e, why does experimental data deviate from the area law for low L? What do simulations predict?*

Our reply: This phenomenon can be understood as due to the finite size effect, as shown in Fig. R2. Here, we perform simulations and tight-binding calculations for the entanglement entropy as a function of the size of the subsystem A. One can see that for small subsystem size, the calculated entanglement entropy indeed deviates from the theoretical scaling laws for both the gapless and gapped phases. From the theoretical perspective, the scaling laws of the entanglement entropy is derived for thermodynamic systems with nearly infinite size. Meanwhile, the subsystem's size is also assumed to be much larger than the lattice constant but much smaller than the size of the entire system. Thus, the experimental results in finite systems can approach the theoretical limit only when the size of the entire phononic crystal is sufficiently large (we use phononic crystals with 40 unit-cells) and the subsystem's size is neither too small nor too large.

Figure R2 | Calculated scaling of the entanglement entropy with the subsystem's size. a, Entanglement entropy versus the subsystem's size for three different cases obtained from finite-element simulations of the acoustic waves. **b**, Entanglement entropy versus the subsystem's size for three different cases obtained from the tight-binding calculations based on the SSH model. In **b**, w and v label the inter- and intra-unit-cell couplings in the SSH model, respectively.

Reviewer's comments: *For low L, which is more practical in experimental settings, can a clear topological transition be predicated, as shown in Fig. 2d?*

Our reply: According to the reviewer’s question, we did simulations and calculations to explore this issue. As shown in Fig. R3, both entanglement entropy and entanglement spectrum can still faithfully indicate the topological phase. However, due to the finite size effect the topological transition becomes less sharp in both entanglement entropy and entanglement spectrum for small subsystem sizes. Nevertheless, for $L \geq 6a$, the entanglement entropy shows a notable peak at the topological transition, while the entanglement spectrum exhibits an abrupt change at the same transition point at $r_1 = 4\text{mm}$. These results indicate that both entanglement entropy and entanglement spectrum can give reliable indication of the topological transition even at small subsystem sizes. We added these results in the revised supplementary information and mentioned it in the main text.

Figure R3 | Topological transition monitored by entanglement entropy and entanglement spectrum at small subsystem sizes. a, Topological transitions monitored by the entanglement entropy for various subsystem sizes. **b,** Topological transitions monitored by the entanglement spectrum for the same cases. All data from finite-element simulations of acoustic waves.

Reviewer’s comments: *How does the location of subsystem A influence the area law shown in Fig. 2e? For example, if one takes the same size of subsystem but closer to the boundary of the chain, would one expect any deviation?*

Our reply: The salient properties of the entanglement entropy studied in this work originate from the properties of the bulk states. If the subsystem is close to the boundary, then the boundary states will come into play together with the bulk states and modify some fine features. To show the impact of boundary states for the area law in Fig. 2e, we perform tight-binding calculations for the entanglement entropy of two partitions--one locates in the middle of the system and the other locates at the boundary of the system (as shown in Fig. R4 below). We find that the location of subsystem A does not impact the overall scaling behavior of entanglement entropy.

The two partition schemes do give different quantitative behaviours. This is mainly because they have different boundaries. For the scheme where the subsystem A is at the middle of the whole system (named as “middle partition”), there are two boundaries of the subsystem A. In contrast, for the other scheme (named as “boundary partition”), there is only one boundary for the subsystem A. For the gapped phases (including the trivial and topological phases), different partitions give the same scaling behavior but quantitatively different entanglement entropy (see Fig. R4). Here, because the nonlocal correlation is weaker for the boundary partition scheme, its entanglement entropy is smaller (meaning that more information is kept in the subsystem, rather than leaking

out via the boundary through nonlocal correlations). In particular, for the topological phase with a large band gap, the two partition schemes give entanglement entropy $\log 2$ and $2\log 2$, respectively. The difference here is because the middle partition scheme has two boundaries and hence according to the area law, doubled entanglement entropy. For the gapless phase, similar reasons lead to quantitatively different scaling. Because there is only one propagating mode at the boundary for the boundary partition scheme, the scaling law becomes $S_A \sim \frac{1}{6} \log L_A$. In contrast, for the middle partition scheme, there are two boundaries of the subsystem A, and the scaling law is $S_A \sim \frac{1}{3} \log L_A$ (keeping in mind that the results obtained from finite systems are valid only when the subsystem size is neither too small nor too large). Nevertheless, all these results are consistent with the theoretical predictions according to Eq. (2) in the main text (they are also consistent with the conformal field theory picture for the gapless phase and the area law for the gapped phase). We added the discussions of these results in the revised supplementary information and mentioned the main results in the main text.

Figure R4 | Entanglement entropy scaling behaviours for different subsystem partitions. Upper panels: illustrating two different subsystem partitions. Lower panels: (left) The scaling behaviours for the two partition schemes for the trivially gapped phase in 1D. (middle) The scaling behaviours for the two partition schemes for the gapless phase in 1D. (right) The scaling behaviours for the two partition schemes for the topologically gapped phase. Calculations are done based on the tight-binding SSH model. Here, w and v label the inter- and intra-unit-cell couplings in the SSH model, respectively.

Reviewer's comments: *Does the value $2\log 2$ for the entanglement entropy depend on the fact the system has chiral or inversion symmetry? Or the existence of edge states is sufficient for one to arrive at this value? If the latter is true, how can one justify that the state is always topologically nontrivial?*

Our reply: This requires no chiral symmetry but only the inversion symmetry, which is the advantage of the entanglement entropy approach. To illustrate this property, we

consider breaking the chiral symmetry by the next-nearest neighboring hoppings in the calculations. In addition, we also consider adding a constant potential on the boundary lattice sites in the SSH model to remove the edge states. In both cases, we show that the entanglement entropy and entanglement spectrum give reliable indication for the topological phases (Note that in these calculations, the inversion symmetry is still preserved as the bulk topology is protected by the inversion symmetry). As shown in the Fig. R5, we demonstrate the features of the SSH model with and without the chiral symmetry in the upper and lower rows, respectively. We observe that the value of $2\log 2$ does not depend on the chiral symmetry in Figs. R5b and R5e. Without the chiral symmetry, the usual bulk-edge correspondence can fail, whereas the entanglement entropy and entanglement spectrum still give the correct indication of the topological phases. Our work indicates that entanglement entropy and entanglement spectrum can be a reliable and direct bulk probe of the bulk topology without relying on the usual bulk-edge correspondence or the edge states. We added these discussions in the revised *Supplementary Note 6* and mentioned the main conclusion in the revised main text.

Figure R5 | SSH model with or without the chiral symmetry. (a) Edge spectrum versus the intra-unit-cell coupling v for the SSH model with chiral symmetry. (b) and (c) Entanglement entropy and entanglement spectrum versus the intra-unit-cell coupling v for the SSH model in (a). (d) Edge spectrum versus the intra-unit-cell coupling v for the modified SSH model without the chiral symmetry. (e) and (f) Entanglement entropy and entanglement spectrum versus the intra-unit-cell coupling v for the modified SSH model in (d), respectively. For all the calculations here, the filling of the valence band is considered. The entire system has 80 unit-cells, while the subsystem has 40 unit-cells. $w = 1$ for all cases. For the modified SSH model in (d), the next-nearest-neighbor hopping is 0.2, and the onsite potential for the two unit-cells at the left and right edge boundaries are 1.

Reviewer's comments: *What explains the region of local minima in Fig. 3d? I believe this depends on the shape of the subsystem A. Therefore, the authors may want to discuss the effect of choosing different shapes of A.*

Our reply: In fact, the local minima in Fig. 3d has nothing to do with the shape of the subsystem A. Instead, the local minima come from the Dirac points. Since Dirac points have a vanishing density of states, when the Fermi level is aligned with the Dirac points, the collective excitations at the Fermi level is suppressed. This leads to the suppression of the nonlocal correlation carried by these collective excitations and thus the reduction of the entanglement entropy. This case corresponds to the minima in the entanglement entropy. Shifting the Fermi level above or below the Dirac points will create a finite-sized Fermi surface and hence the enhanced nonlocal correlations and increased entanglement entropy. To numerically confirm this picture, we simulate the phononic entanglement entropy for three shapes of the subsystem A as illustrated in Fig. R6. The results show that for distinct shapes of the subsystem A, the local minimum in the entanglement entropy is pinned at the situation when the Fermi level is at the Dirac points, i.e., $\mu = 0$. We added discussions on this issue in the revised supplementary information and mentioned the main conclusion in the revised main text.

Figure R6 | Entanglement entropy for different subsystem partitions in 2D honeycomb lattice. (Left) Schematic illustration of different shapes of the subsystem A: parallelogram, rectangle, triangle, and rhombus. (Right) The non-universal coefficient of the leading term of entanglement entropy versus the chemical potential μ for different subsystem partitions in the 2D honeycomb lattice, where the local minimum of the non-universal coefficient is unchanged for the different shapes of the subsystem A. All results are obtained from the tight-binding calculations based on the nearest-neighbor honeycomb lattice model. Here the entire system has 80×80 unit-cells. The linear length of the subsystems A is $L_A = 20$ for each partition scheme.

Reviewer's comments: *At what frequency Figs. 4c and 4e are plotted? I believe the edge spectrum has nonzero group velocity, shown in Fig. 4d. Therefore, it is not clear how the choice of frequency would influence the entanglement spectrum and entropy and prove the existence of edge states.*

Our reply: For the integration of the correlation function in Fig. 4, the lower frequency is set as 6.67kHz (41.9krad/s), while the upper frequency is chosen as the frequency of the Dirac points, i.e., 7.62kHz (47.9krad/s). Such a setting is because we are considering the analog of free fermions with the valence band entirely filled. We emphasize here again that entanglement entropy and entanglement spectrum characterize only the bulk properties and do not depend on the edge states (as also shown by the simulations and calculations above). Therefore, the edge states do not affect the measured entanglement entropy and entanglement spectrum. In addition, we remark that entanglement entropy and entanglement spectrum do not give evidence for the existence of the edge states but give a direct characterization of the bulk topology. Here, the emergence of the edge states depends on whether the system has the chiral symmetry. If the system has the chiral symmetry, then the conventional bulk-edge correspondence is valid, and the edge states emerge. Without the chiral symmetry, the edge states can be annihilated even if the bulk topology is nontrivial. Nevertheless, in both cases, the nontrivial bulk topology can be faithfully characterized by the entanglement entropy and entanglement spectrum as elaborated above.

Reviewer's comments: - *The colormap in Fig. 4d needs an explanation.*

Our reply: The colormap of Fig. 4d gives the intensity of the detected acoustic signal after Fourier transformation for the pump-probe measurements along an edge boundary. That is, it gives the spectral intensity along the edge boundary at various wavevectors and frequencies. We add some explanations in the figure caption.

REVIEWERS' COMMENTS

Reviewer #1 (Remarks to the Author):

Dear Editors,

I feel that the authors have addressed in a satisfactory manner the requests of all the referees. I recommend publication in Nature Communications.

Reviewer #3 (Remarks to the Author):

The authors have addressed all my concerns. I recommend publication as is.